# Stored Intestinal Biopsies in Inflammatory Bowel Disease Research: A Danish Nationwide Population-Based Register Study

**DOI:** 10.3390/jpm15040129

**Published:** 2025-03-28

**Authors:** Heidi Lynge Søfelt, Jessica Pingel, Donna Lykke Wolff, Karen Mai Møllegaard, Silja Hvid Overgaard, Anders Green, Gunvor Iben Madsen, Niels Qvist, Sofie Ronja Petersen, Trine Andresen, Andre Franke, Niels Marcussen, Robin Christensen, Vibeke Andersen

**Affiliations:** 1Molecular Diagnostics and Clinical Research Unit, Department of Internal Medicine, University Hospital of Southern Denmark, 6200 Aabenraa, Denmark; heidi.lynge.sofelt@rsyd.dk (H.L.S.); jpingel@health.sdu.dk (J.P.); kmai@rsyd.dk (K.M.M.); siljahvid@msn.com (S.H.O.); trinea@hst.aau.dk (T.A.); 2Winsløw Unit for Anatomy, Histology and Plastination, Department of Molecular Medicine, University of Southern Denmark, 5000 Odense, Denmark; 3Internal Medicine Research Unit, Department of Internal Medicine, University Hospital of Southern Denmark, 6200 Aabenraa, Denmark; donna.wolff@rsyd.dk; 4Department of Clinical Research, University Hospital of Southern Denmark, 6200 Aabenraa, Denmark; srpe@rsyd.dk; 5Institute of Regional Health Research, University of Southern Denmark, 5000 Odense, Denmark; niels.marcussen@rsyd.dk; 6Section for Biostatistics and Evidence-Based Research, The Parker Institute, Bispebjerg and Frederiksberg Hospital, University of Copenhagen, 2000 Copenhagen, Denmark; robin.christensen@regionh.dk; 7Steno Diabetes Center Odense, Odense University Hospital, 5000 Odense, Denmark; agreen@dadlnet.dk; 8Department of Clinical Research, University of Southern Denmark, 5000 Odense, Denmark; niels.qvist@rsyd.dk; 9Department of Clinical Pathology, Odense University Hospital, 5000 Odense, Denmark; 10Research Unit for Surgery and Centre of Excellence in Gastrointestinal Diseases and Malformations in Infancy and Childhood (GAIN), Odense University Hospital, 5000 Odense, Denmark; 11Institute of Clinical Molecular Biology, Christian-Albrechts-University of Kiel, 24118 Kiel, Germany; a.franke@ikmb.uni-kiel.de; 12University Medical Hospital Schleswig Holstein, Christian-Albrechts-University of Kiel, 24098 Kiel, Germany; 13Department of Clinical Pathology, University Hospital of Southern Denmark, 6200 Aabenraa, Denmark; 14Research Unit of Rheumatology, Department of Clinical Research, University of Southern Denmark, Odense University Hospital, 5000 Odense, Denmark; 15Institute of Molecular Medicine, University of Southern Denmark, 5000 Odense, Denmark

**Keywords:** biopsies, inflammatory bowel disease, ulcerative colitis, Crohn’s disease, formalin-fixed paraffin-embedded samples, gastrointestinal

## Abstract

**Background.** Inflammatory bowel disease (IBD), encompassing Crohn’s disease (CD) and ulcerative colitis (UC), is a complex inflammatory condition affecting the intestinal tract. Currently, immune-modulating treatments are inadequate for 30–50% of patients and often cause significant side effects, highlighting the urgent need for a personalized medicine approach. Real-world data and archived gut biological material from clinical repositories could be a resource for identifying new drug candidates and biomarkers. This study assesses the extent of stored formalin-fixed, paraffin-embedded (FFPE) gut biopsies from patients with IBD that could be leveraged for research efforts. **Methods**. Data from the Danish National Patient Register and the Danish Pathology Register were used to construct a cohort of patients diagnosed with IBD between 1 January 2005, and 30 June 2013, and followed for five years. **Results**. Among 14,512 IBD patients, 13,936 (96%) had at least one biopsy visit within five years after their initial diagnosis (CD 94%, UC 97%), and 13,598 (94%) had their first biopsy visit as part of the diagnostic process. Biopsies were taken from the colon (82%) or multiple locations (46%). Patients with severe disease had more biopsy visits than those with non-severe disease (IBD 3.3 vs. 2.0 visits, CD 2.9 vs. 1.9 visits, UC 3.6 vs. 2.0 visits). **Conclusions**. Thus, the vast majority of patients with IBD have biopsies taken. These findings demonstrate the feasibility and applicability of combining real-world data and archived gut biopsies for research, highlighting it as a valuable but underutilized resource for identifying new drug candidates and biomarkers, with huge potential for enhancing personalized medicine within IBD for the benefit of patients and society.

## 1. Introduction

Inflammatory bowel disease (IBD), including Crohn’s disease (CD) and ulcerative colitis (UC), is a chronic gastrointestinal disorder [1,2,3]. It affects nearly 1% of the Western population, which makes it a significant public health challenge [4,5]. CD and UC present with relapsing symptoms and varied treatment responses. Worryingly, 30–50% of patients experience insufficient treatment outcomes, severe side effects, or loss of response, and a similar proportion require surgery within five years of diagnosis [6,7,8]. This underscores the urgent need for better disease management.

Personalized medicine offers a path forward by optimizing existing treatments and developing reliable diagnostic and prognostic biomarkers. Biomarkers can improve diagnosis and predict treatment responses, enabling more precise, individualized therapy [9]. For example, diagnostic biomarkers enable precise diagnoses, forming the foundation for targeted drug treatments, and predictive biomarkers guide the selection of the optimal drug for each patient. Advancing this approach requires a deep understanding of disease mechanisms, identification of therapeutic targets, and validation of biomarker candidates [9,10].

IBD research relies on large patient cohorts with detailed clinical data and biological samples, ideally from the gut, where diet, microbes, and metabolites drive inflammation [11]. However, obtaining gut tissue is challenging due to invasive procedures, patient risks, high costs, and resource demands. As a result, many clinical decisions rely on limited evidence [12]. Moreover, despite phase 3 clinical studies being the gold standard, their short follow-up (1–2 years) may introduce bias, and prospective biomarker identification has shown limited success.

Real-world data and archived biopsies from clinical repositories offer a potential solution. Retrospective analysis of archived gut biopsies, with long-term follow-up data, could help identify pharmacologic targets and biomarkers for personalized medicine. However, despite the potential of biopsy archives from IBD patients followed for decades, they remain underutilized. Importantly, to fully harness their potential, it is essential to integrate molecular biopsy analyses with register-based detailed clinical data on when and why the biopsies were taken. This combined approach could enhance IBD research and support more precise, evidence-based treatment strategies.

This study will use Danish National Registry data to: (i) determine the proportion of IBD patients with biopsies, (ii) quantify biopsy visits, (iii) map the gastrointestinal locations of biopsies, and (iv) assess the association of disease severity with the number of biopsies. By analyzing the availability and characteristics of stored pathological samples, this study will highlight the potential of archived biopsies for advancing personalized medicine research in IBD.

## 2. Methods

### 2.1. Study Design and Setting

This study is a historic register-based cohort study investigating patients with IBD.

### 2.2. Data Sources

All data was retrieved from the Danish Civil Registration System (CRS), the Danish National Patient Register (DNPR), and the Danish Pathology Register (DPR). In Denmark, all civil data for Danish citizens, including birth, death, and migration dates, have been registered in CRS since 1968 [13]. A unique 10-digit identification number (CPR number) is assigned to all Danish citizens at birth or immigration, providing opportunities for person-level linkage between Danish health registers [13,14]. DNPR was established in 1977 and contains all information about inpatient and outpatient contacts (outpatient since 1995) with Danish hospitals [15]. Information on gastrointestinal biopsies was retrieved from DPR, a register established in 1997 by the National Board of Health providing information on pathological examinations performed at pathology departments [16,17].

### 2.3. Study Population

This nationwide register-based cohort study comprises all Danish citizens (i.e., children and adults) with an incident IBD diagnosis from 1 January 2005 to 30 June 2013. IBD diagnosis was defined as at least two registrations of IBD in the inclusion period (1 January 2005–30 June 2013) according to DNPR [15]. IBD was defined according to the 10th revision of the International Classification of Diseases (ICD10) diagnostic codes K50 (incl. all subcodes) for CD and K51 (incl. all subcodes) for UC.

Patients were followed for five years, and those lost during the follow-up period due to death or emigration in this period were excluded. Furthermore, patients were excluded if they had a previous registration of IBD during the “wash out” period from 1977 until 31 December 2004, to ensure incident cases. Therefore, the registration with the former ICD-8 codes for IBD (563.01, 563.02, 563.08, 563.09, 563.19, 569.04) as well as ICD-10 (K50 and K51) was used as an exclusion in the “wash out” period. Primary (A-diagnoses) and secondary diagnoses (B-diagnoses) were accepted.

IBD subgroups (i.e., CD and UC) were determined based on the second registration of IBD. The date of study entry (index date) was defined as the first registration of IBD.

### 2.4. Variables

The primary outcome of this study was to investigate the proportion of IBD patients (CD and UC, respectively) who had biopsies taken one year or before five years after their first registration for IBD. Pathology evaluation of intestinal biopsies is often performed as a part of diagnosing IBD and, therefore, occurs prior to the first registration of IBD. Thus, visits with biopsies taken in the past year before the first registration were also included.

A biopsy visit was defined as a visit with biopsies taken, regardless of the quantity of biopsies. The number of biopsies taken at each visit may vary according to clinical practice and diagnostic findings. The first biopsy visit was defined as the first biopsy visit in the period from one year before until five years after the first registration of IBD. The period from one year before until one year after the first registration of IBD will be referred to as the diagnostic process.

Information on gastrointestinal biopsies was retrieved from DPR using gastrointestinal topography codes and used to categorize the gastrointestinal location of biopsies [16,17]. For each biopsy visit, the gastrointestinal location from where the biopsies were taken was categorized as either the upper gastrointestinal tract (T62, T63, all subcodes), small intestine (T64, T65, all subcodes), colon (T66, T67, T68, all subcodes and T69000, T69010), or multiple locations (a combination of the three locations).

Severe disease course was defined as a composite of treatment with biologics, or IBD-related hospitalization for ≥7 days, or IBD-related intestinal surgery during follow-up. Treatment with biologics encompasses one of the following codes: BOHJ18A (TNFi), BOHJ18B and L04AC (Interleukin inhibitors), BOHJ19 (Vedolizumab), or BOHJ28 (Tofacitinib). IBD-related hospitalization was defined as hospitalization for ≥7 days, with IBD diagnostic codes K50 or K51 as the primary diagnosis code. IBD-related surgery was defined as intestinal surgery, i.e., one of the following codes for CD: KJFB00-97 (except KJFB10-13), KJGB00-97 (except KJGB20), and KJFH00-21, and for UC: KJFH10-21, and KJFH30-34, and KJFH96 [18].

Co-variables were chosen based on their possible influence on the risk of having a severe disease course over time and/or the association between having a severe disease course and the number of biopsy visits. Information on the patient’s age and sex was included. The age of the patient was recorded when IBD was diagnosed.

### 2.5. Statistical Analysis

In this study, descriptive statistics were used to summarize aggregated participant characteristics collected over the entire five-year period for all patients, including CD and UC. Categorical and dichotomous variables were reported as numbers and percentages (*n*, %), and continuous variables as medians and interquartile ranges (IQR). Differences between the two groups (CD and UC) and between severe and non-severe disease courses were tested using Pearson’s chi-squared test (categorical variables), as the expected number of observations was ≥5 or the Wilcoxon rank-sum test (continuous variables), depending on the data type.

A sensitivity analysis was performed to test the robustness of the present results, and a stratified analysis was performed according to diagnosis to account for differences in characteristics between groups (CD and UC). Statistical programming and analyses were performed using the software STATA version 17. All graphs were created in Graph Pad Prism version 6.0. *p*-values of <0.05 were considered statistically significant.

## 3. Results

### 3.1. Patient Characteristics

Of the included patients, 32% were diagnosed with CD and 68% with UC (Figure 1 and Table 1).

### 3.2. Proportion of Patients Having Biopsies Taken and Number of Biopsies

Many patients had biopsies taken in the study period (IBD: 96%, CD: 94%, and UC: 97%) (Table 1). Most patients had their first biopsy visit during the period of the diagnostic process (IBD: 94%, CD: 92%, and UC: 95%). Approximately two-thirds of the patients in the study population had at least two biopsy visits (IBD: 66%, CD: 67%, and UC: 65%), and more than a third had three or more biopsy visits during follow-up (IBD: 39%, CD: 40%, UC: 39%) (Table 1).

### 3.3. Gastrointestinal Location of Biopsies

Biopsies were most often taken in the colon (82%). Patients finally diagnosed with CD: 61% or UC: 88% had a biopsy visit with biopsies taken only in the colon during follow-up. The second most frequent gastrointestinal location was “multiple locations” IBD: (46%), CD: (68%), and UC: (38%). Furthermore, 12% of all patients had biopsies taken only from the small intestines (CD: 23%) and (UC: 6%). Finally, 4% of all IBD patients (CD: 5%, UC 3%) had biopsies taken from the upper gastrointestinal tract (Table 1).

### 3.4. Severe Disease Course and Number of Biopsy Visits

In total, 37% of all patients had a severe disease course during the follow-up period defined by having experienced at least one of the following situations: treatment with biologics, IBD-related hospitalization for ≥7 days, or IBD-related intestinal surgery during follow-up. This was more prevalent among patients diagnosed with CD (54%) than patients diagnosed with UC (29%). This also applied to individuals experiencing a severe disease course due to initiating treatment with biologics (CD: 32%, UC: 16%), IBD-related hospitalization ≥ 7 days (CD: 28%, UC: 21%), or IBD-related surgery (CD: 21%, UC: 8%) (Table 1).

Patients with IBD experiencing a severe disease course had more biopsy visits compared to those having a non-severe disease course during follow-up (Figure 2A–C). Accordingly, patients with IBD and a severe disease course had 3.3 biopsy visits; in contrast, patients with a non-severe disease course had 2.0 biopsy visits (CD: 2.9 vs. 1.9 visits, UC: 3.6 vs. 2.0 visits).

## 4. Discussion

This observational register-based cohort study observed that 96% of the patients with IBD had at least one biopsy visit, and 23% had more than three biopsy visits within 5 years from diagnosis (Table 1). Further, 94% of the patients had a biopsy visit during the diagnostic process. Biopsies mainly were taken from the colon (82%) or multiple locations (46%). In addition, IBD patients with a severe disease course had more biopsy visits compared to those having a non-severe disease course (3.3 vs. 2.0 visits).

### 4.1. Comparison with Other Similar Studies

We were not able to find studies combining register-based clinical information with molecular analysis of archived biopsies. The fact that the vast majority of patients with IBD have biopsies taken underscores the huge research potential of biopsies from clinical repositories. Our study is unique in highlighting this unutilized potential for advancing personalized medicine within IBD.

Biopsies are often taken during IBD diagnosis and management and are typically stored as formalin-fixed, paraffin-embedded (FFPE) samples in repositories. Diagnostic biopsies, collected before treatment initiation, could be well-suited for identifying individual patients’ baseline inflammatory profiles, and follow-up biopsies may enable the study of inflammatory changes over time, reflecting disease progression and treatment effects [19,20]. In line with this, new advanced molecular methods for FFPE biopsies are emerging [21,22,23,24,25,26]. For instance, FFPE tumor samples have been used to understand cancer biology [27]. The current study thus highlights the potential of combining real-world data and archived biopsies from clinical repositories, offering unique insights into disease mechanisms, new drug targets, and biomarker identification.

### 4.2. Strengths and Limitations of the Study

The main strength of this study is the use of person-level data linked between health registers in a large nationwide population-based cohort, which reduced the risk of selection bias. Another main strength is the long follow-up time. Danish register data has a high completeness and validity, and the use of real-world data increases the generalizability of the present results [13,15,28]. As another strength, patients with at least two IBD registrations in DNPR were included. However, the present study also has some limitations, including a lack of data on additional covariates, which could have had a confounding effect on the results, for instance, smoking, a known risk factor associated with both CD and UC [2,29]. Furthermore, another potential limitation is related to the classification of patients as either having CD or UC. Studies have observed that the distinction between the two diagnoses is sometimes difficult, mainly due to the heterogeneous presentation of clinical phenotypes [3,20]. Consequently, the diagnosis was changed for 9% of the patients, from either CD to UC or vice versa, after the initial diagnosis had been made [3]. Therefore, to minimize the risk of misclassification, the second registration of IBD was chosen to classify the diagnosis as either CD or UC [3]. Finally, registry studies detect associations but cannot detect causality.

## 5. Conclusions

In summary, the vast majority of patients with IBD have biopsies taken. These findings demonstrate the feasibility and applicability of combining real-world data and archived gut biopsies for research, highlighting it as a powerful yet underutilized resource for identifying new drug targets and biomarkers. This approach holds immense potential to advance personalized medicine in IBD, enabling more precise diagnosis, prognosis, and treatment selection, benefitting both patients and society.

## Figures and Tables

**Figure 1 jpm-15-00129-f001:**
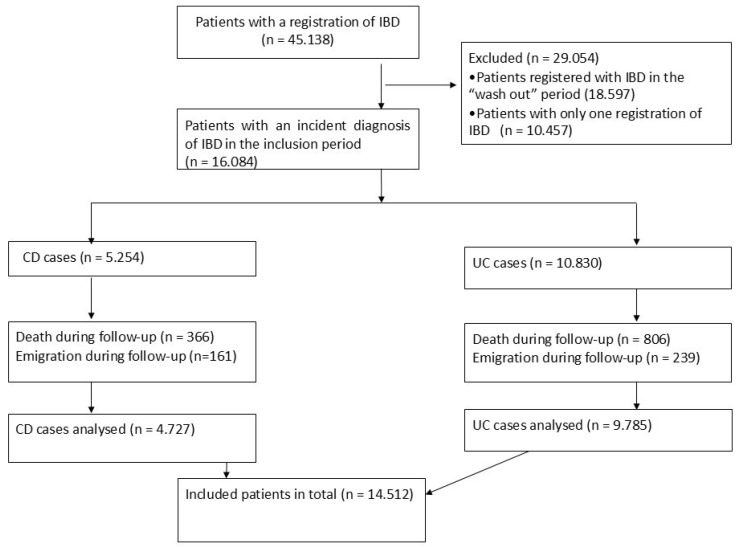
Inflammatory bowel disease (IBD); Crohn’s disease (CD); ulcerative colitis (UC); inclusion period: January 2005 to June 2013.

**Figure 2 jpm-15-00129-f002:**
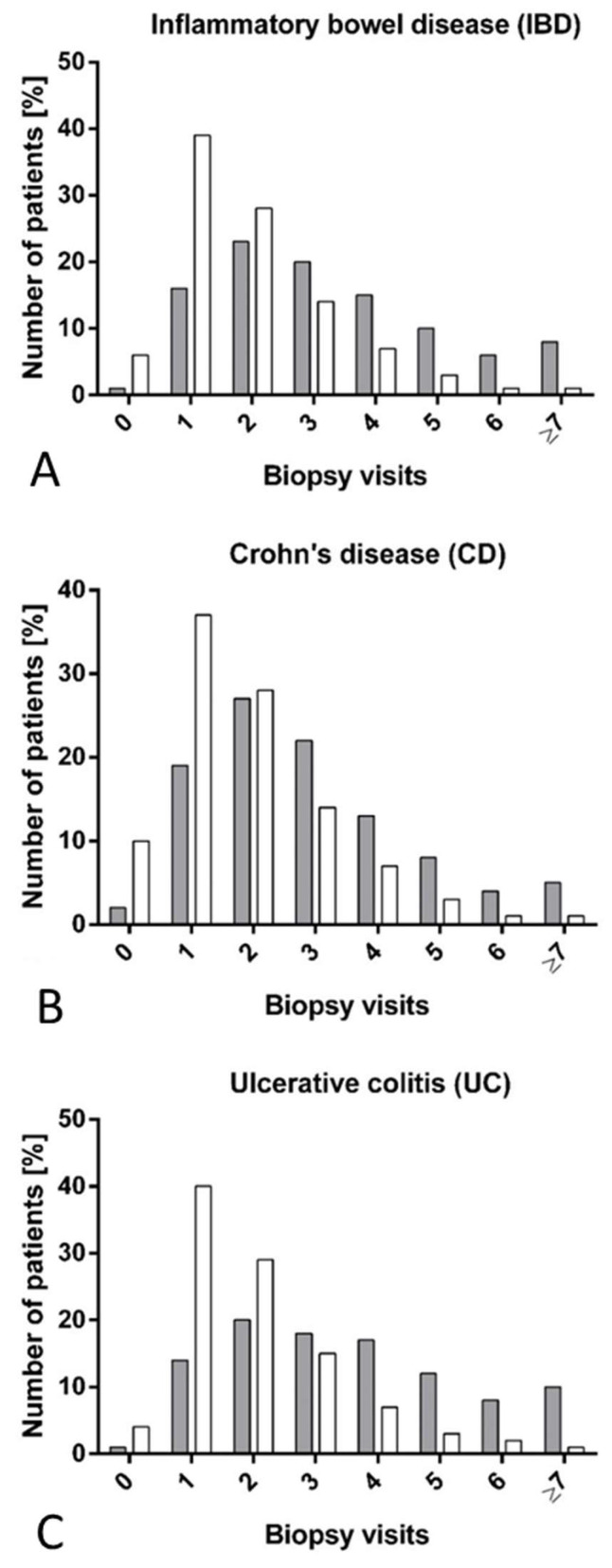
Biopsy visits among patients having a severe disease course and patients having a non-severe disease course. The grey bars represent the patients with severe disease and the open bars those without. (**A**): Biopsy visits among patients diagnosed with IBD. (**B**): Biopsy visits among patients diagnosed with Crohn’s disease (CD). (**C**): Biopsy visits among patients diagnosed with ulcerative colitis (UC). Severe disease course is a composite outcome defined as treatment with biologics or IBD-related hospitalization for ≥7 days or IBD-related intestinal surgery. Number of biopsy visits consists of biopsy visits in the period one year before until five years after the first registration of IBD.

**Table 1 jpm-15-00129-t001:** Participant characteristics.

	IBD	CD	UC	*p*-Value
Enrolled participants, *n* (%)	14,512	4727 (32)	9785 (68)	
Females, *n* (%)	7675 (53)	2605 (55)	5070 (52)	<0.001
Age at IBD diagnosis (median, IQR)	38 (25–54)	32 (21–49)	40 (27–56)	<0.001
Patients with at least one biopsy visit, *n* (%)	13,936 (96)	4463 (94)	9473 (97)	<0.001
Patients having their first biopsy visit				<0.001
Patients with no biopsy visit, *n* (%)	576 (4)	264 (6)	312 (3)	
In the period of diagnostic process, *n* (%)	13,598 (94)	4329 (92)	9269 (95)	
>1 ≤2 years after their first registration of IBD, *n* (%)	144 (1)	59 (1)	85 (<1)	
>2 ≤3 years after their first registration of IBD, *n* (%))	94 (<1)	40 (<1)	54 (<1)	
>3 ≤4 years after their first registration of IBD, *n* (%)	53 (<1)	15 (<1)	38 (<1)	
>4 ≤5 years after their first registration of IBD, *n* (%)	47 (<1)	20 (<1)	27 (<1)	
Accumulated biopsy visits				<0.001
0, *n* (%)	576 (4)	264 (6)	312 (3)	
1, *n* (%)	4452 (31)	1295 (27)	3157 (32)	
2, *n* (%)	3843 (27)	1294 (27)	2549 (26)	
3, *n* (%)	2381 (16)	849 (18)	1532 (16)	
>3, *n* (%)	3260 (23)	1025 (22)	2235 (23)	
Patients with biopsies from different gastrointestinal locations †				<0.001
Upper gastrointestinal tract, *n* (%)	579 (4)	244 (5)	335 (3)	
Small intestine, *n* (%)	1725 (12)	1107 (23)	618 (6)	
Colon, *n* (%)	11,463 (82)	2863 (61)	8600 (88)	
Multiple locations, *n* (%)	6668 (46)	3051 (65)	3617 (37)	
Patients with non-severe disease course	9136 (63)	2195 (46)	6941 (71)	<0.001
Patients with severe disease course, yes, *n* (%)	5376 (37)	2532 (54)	2844 (29)	<0.001
Patients with severe disease course because of: †				
Initiating biologics, *n* (%)	3059 (21)	1518 (32)	1541 (16)	<0.001
IBD related hospitalization ≥7 days, *n* (%)	3410 (24)	1342 (28)	2068 (21)	<0.001
IBD related surgery, *n* (%)	1747 (12)	988 (21)	759 (8)	<0.001

IBD: inflammatory bowel disease, CD: Crohn’s disease, UC: ulcerative colitis, *n*: number of patients. Biopsy visit is defined as a visit with biopsies taken, regardless of the number of biopsies taken. Time of first biopsy visit is defined as the first biopsy visit in the period one year before until five years after the first registration of IBD. Period of the diagnostic process refers to the period of one year before until one year after the first registration of IBD. Severe disease course is a composite outcome defined as treatment with biologics or IBD-related hospitalization for ≥7 days or IBD-related intestinal surgery, IQR is interquartile range. † Patients can appear in more than one category; patients are included in a category if they had at least one biopsy visit with biopsies taken from the gastrointestinal location in question in the period one year before until one year after their first registration of IBD. The difference between the two groups (CD and UC) was tested using Pearson’s chi-squared test or Wilcoxon rank-sum test.

## Data Availability

The Danish Health Data Authority (Sundhedsdatastyrelsen) and the Agency of Statistics Denmark (Danmarks Statistik) provided the data used for this study. All data was used with permission from Danmarks Statistik. Data can be shared by request to the corresponding author but requires permission from Danmarks Statistik.

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
