# Peer review of "Stored Intestinal Biopsies in Inflammatory Bowel Disease Research: A Danish Nationwide Population-Based Register Study"

_jpm, 2025, doi:10.3390/jpm15040129_

Round 1

Reviewer 1 Report

Comments and Suggestions for Authors

The manuscript is well organized and clearly expressed. The structure of the article is relatively complete and novel. However, there are still several issues worth considering:

  1. Most of the text uses the abbreviation "IBD", but in some places "inflammatory bowel disease" is used instead. To avoid confusion, a uniform format is recommended.
  2. The introduction is comprehensive, but the wording could be adjusted slightly to make it more concise and academic. For example, "assess whether a severe disease course is associated with the number of biopsy visits " could be changed to "assess the association of disease severity with the number of biopsies."
  3. The description of statistical analysis is relatively basic. It is suggested to briefly explain the reasons for choosing Pearson Chi-square test and Wilcoxon rank sum test, and which study variables were used respectively.
  4. In the discussion section, the summary of the research results is slightly direct, and it is suggested to add a comparison with other similar studies to illustrate the unique contribution of this study.

Author Response

Reviewer 1

The manuscript is well organized and clearly expressed. The structure of the article is relatively complete and novel. However, there are still several issues worth considering:

  1. Most of the text uses the abbreviation "IBD", but in some places "inflammatory bowel disease" is used instead. To avoid confusion, a uniform format is recommended.
  2. The introduction is comprehensive, but the wording could be adjusted slightly to make it more concise and academic. For example, "assess whether a severe disease course is associated with the number of biopsy visits " could be changed to "assess the association of disease severity with the number of biopsies."
  3. The description of statistical analysis is relatively basic. It is suggested to briefly explain the reasons for choosing Pearson Chi-square test and Wilcoxon rank sum test, and which study variables were used respectively.
  4. In the discussion section, the summary of the research results is slightly direct, and it is suggested to add a comparison with other similar studies to illustrate the unique contribution of this study.

Reply R1

Thank you to the reviewer. The comments have helped us to improve the manuscript, in particular the language.

  1. I have been through the text but found no places where IBD was not abbreviated, apart from the explanation in tables/figures.
  2. Thank you for noticing us. We have shortened the introduction and improved the language. Changes in the introduction are marked by yellow.
  3. Thank you for the remark. We have clarified the applied tests in the respective paragraph.
  4. Thank you for this comment. The reason was that no other studies were found. We have now made this clear and write: “Comparison with other similar studies

We were not able to find studies combining register-based clinical information with molecular analysis of archived biopsies. The fact that the vast majority of patients with IBD have biopsies taken underscores the huge research potential of biopsies from clinical repositories. Our study is unique in highlighting this unutilised potential for advancing personalised medicine within IBD. “

Reviewer 2 Report

Comments and Suggestions for Authors

Søfelt et al took advantage of a nationwide registry to underpin the importance of an underused tool in exploring novel treatments for IBD. They found that, for 94% of patients, biopsies were part of the initial diagnosis, that over 95% of them had more sets of biopsies within the next 5 years and that biopsies were taken more often in severe disease.

Retrospectively, going back to archives of biopsies of people with an already known disease course over long follow-up periods could serve as a tool to identify pharmacologic targets in the era of personalized medicine. Phase 3 clinical studies are the gold-standard but, inevitably, these are short-term (1-2 years) studies and biomarkers identified prospectively seem to be of low or no use so far.

This might be a biased conclusion due to short follow-up time. There might be a wealth of evidence in the biopsy archives of IBD patients followed up for decades, but this is currently underutilized or not at all utilized.

The main weak point of this study is that it lacks clinical or in silico or wet-lab novelty. Narrative data are presented on the obvious that patients with severe disease undergo endoscopy and biopsies more often.

Specific points

Page 2, “Notably, … taken”: This seems like an overstatement. Intestinal loci biopsies are taken form are routinely noted at the endoscopy reports and if one goes back to the clinical records of the outpatient visits, hospitalizations and biologic infusions to an IBD patient, the information is there,

Page 8, strong points of the study: Please discuss long observation time.

Comments on the Quality of English Language

Page 5, Table 1; Page 6: “Small intestines” -> “Small intestine”

Page 5, Table 1, Footnotes: “the quantity of biopsies taken” -> “the number of biopsies taken”

Page 5, Table 1, Footnotes: use uniformly either “:” or “is”

Page 5, Table 1, Footnotes: “represented” -> “included”

Page 7, Figure 2, Caption: “the severe patients and the open bars represent the non-severe patients” -> “patients with severe disease and open bars those without”

Author Response

Reviewer 2

Søfelt et al took advantage of a nationwide registry to underpin the importance of an underused tool in exploring novel treatments for IBD. They found that, for 94% of patients, biopsies were part of the initial diagnosis, that over 95% of them had more sets of biopsies within the next 5 years and that biopsies were taken more often in severe disease.

Retrospectively, going back to archives of biopsies of people with an already known disease course over long follow-up periods could serve as a tool to identify pharmacologic targets in the era of personalized medicine. Phase 3 clinical studies are the gold-standard but, inevitably, these are short-term (1-2 years) studies and biomarkers identified prospectively seem to be of low or no use so far.

This might be a biased conclusion due to short follow-up time. There might be a wealth of evidence in the biopsy archives of IBD patients followed up for decades, but this is currently underutilized or not at all utilized.

The main weak point of this study is that it lacks clinical or in silico or wet-lab novelty. Narrative data are presented on the obvious that patients with severe disease undergo endoscopy and biopsies more often.

Specific points

Page 2, “Notably, … taken”: This seems like an overstatement. Intestinal loci biopsies are taken form are routinely noted at the endoscopy reports and if one goes back to the clinical records of the outpatient visits, hospitalizations and biologic infusions to an IBD patient, the information is there,

Page 8, strong points of the study: Please discuss long observation time.

Comments on the Quality of English Language

Page 5, Table 1; Page 6: “Small intestines” -> “Small intestine”

Page 5, Table 1, Footnotes: “the quantity of biopsies taken” -> “the number of biopsies taken”

Page 5, Table 1, Footnotes: use uniformly either “:” or “is”

Page 5, Table 1, Footnotes: “represented” -> “included”

Page 7, Figure 2, Caption: “the severe patients and the open bars represent the non-severe patients” -> “patients with severe disease and open bars those without”

Submission Date

20 January 2025

Date of this review

02 Feb 2025 13:46:55

Reply R2

Thank you to the reviewer for the comments, which have helped us to improve the manuscript.

  1. Thank you for the notification of the impact of the follow-up time. We have included this argument in the introduction.
  2. “Notably, the full realisation of the potential of research combining real-world data and archived biopsies needs epidemiological characterisation of the patients with biopsies and information on when (the clinical situations) and where (gastrointestinal locations) biopsies are taken. ” Thank you for this comment. However, to access this information, one needs access to the individual patient record, which permission is not routinely given. Instead, using register information can be a way forward. The sentence has been changed to: “However, to fully harness their potential, it is essential to integrate molecular biopsy analyses with register-based detailed clinical data on when and why the biopsies were taken. This combined approach could enhance IBD research and support more precise, evidence-based treatment strategies.”
  3. Thanks for notifying us. We have added: “Another main strength is the long follow-up time.”
  4. “s” has been deleted.
  5. “Number” has been inserted.
  6. Now “:” is used between abbreviations. Definitions are explained.
  7. “Included” is now used.
  8. Thank you for capturing this mistake. Now “patients with severe disease and open bars those without” is used.